# Innovative Integrated Solution for Monitoring and Protection of Power Supply System from Railway Infrastructure

**DOI:** 10.3390/s21237858

**Published:** 2021-11-25

**Authors:** Mihai Andrusca, Maricel Adam, Alin Dragomir, Eduard Lunca

**Affiliations:** Faculty of Electrical Engineering, Gheorghe Asachi Technical University of Iasi, Bd. Dimitrie Mangeron nr. 21–23, 700050 Iaşi, Romania; adamm@tuiasi.ro (M.A.); elunca@tuiasi.ro (E.L.)

**Keywords:** condition monitoring, fault diagnosis, protection, railway infrastructure, monitoring device, GSM communication

## Abstract

This paper describes an innovative integrated solution for monitoring and protection of the power supply system of electric traction. The development of electronics devices, new possibilities to communicate (wireless), and new sensors makes it possible to design, develop and implement new hardware–software structures in various fields such as energy systems, transportation infrastructure, etc. This contributes to increasing developments in the monitoring and protection of railway infrastructure. A monitoring and protection system that uses sensors and devices to acquire electrical parameters from railway infrastructure has been developed and applied for fault detection and protection of power supply systems from electric traction. The solution of monitoring and protection presented are composed of a hardware–software structure with Global System for Mobile Communications (GSM) communication for monitoring of power supply installations from the electric traction and a central remote system composed of a device with GSM communication and a server that will allow, among others things, accurate detection of the block section (SC), in which an electrical fault (short circuit) has occurred, determination of the circuit breakers electro-erosion from the railway installations and an indication of the opportune moment for maintenance activity, respectively, as well as knowledge of the technical condition of some equipment from the return circuit. The proposed and developed method for monitoring devices has been validated in the railway laboratory to confirm its capability to detect defects and was tested in the field. Experimental results in the field and appropriate data analysis are included in this article.

## 1. Introduction

The paper focuses on an interesting topic on both the international and national level, namely the development, modernization and interconnection of railway infrastructure for passenger and freight transport between countries.

At the European level, among the issues of interest in the field of transport, which are included in the European strategic plan, the focus is on achieving a European transport system that is green, environmentally friendly (European Green Deal), intelligent and innovative through automatization and digitalization of this sector (a Europe fit for the digital age), and that is interconnected, efficient, accessible to all member countries of the European Union (an economy works for people), with a high level of safety and security, and ready for new international challenges (promoting our European way of life) [1].

At the national level, in Romania, according to the railway infrastructure development strategy, the objectives in the railway sector include: Increasing the attractiveness of railway transport, (especially the electric transport, which is more environmentally friendly, having a lower impact on the environment and climate change, compared to the diesel transport), improving travel conditions by modernizing the national railway network and developing railway services, including the automatization and digitalization of this system [2,3].

Transportation on the railway systems is the most reliable, efficient, cost effective and convenient mode of transport. It has minimum fuel costs, is capable of transporting large freights, is environmentally friendly and, most importantly, is also very safe. Railway transport is one of the main ways to move in the world. An efficient transport system is a fundamental condition for prosperity and wellbeing at national and international levels. It is capable of transporting many freights, in an environmentally friendly mode, and is very safe and needs few fuel costs [4]. Railway transport is one of the main ways to move in the world. An efficient transport system is a fundamental condition for prosperity and wellbeing at national and international levels.

However, railway infrastructure management is essential for continuous operation of railway systems. An important component of the management is railway condition monitoring and protection that detects the faults and technical condition degradation of electrical equipment from railway infrastructure concerning various factors, including the weight of railway vehicles on railway tracks, occurrence of short circuits, materials used in railway track infrastructure (steel on electrical connections between impedance bond railway tracks, etc.) and environmental conditions. The aim of railway condition monitoring and protection is to detect the incipient deterioration of some elements from railway infrastructure before causing any damage or preventing railway operations [4,5,6].

In the last few years, there has been a tendency to use railway infrastructure equipment up to their limits, assuming the risk of failure, which leads to power supply interruption of the railway system. Those things lead to undesirable aspects in terms of train circulation (injuries, delays, etc.), with economic and technical losses. Timely detection and identification of faults in railway infrastructure are very important for the availability, reliability and safety of the railway system. In order to reduce the failure risk in railway infrastructure, it is necessary to acquire, collect and process some parameters which give the technical condition of equipment and, thereby, enable prediction of any damage to the electrical equipment [5,7].

Currently, from the point of view of railway infrastructure monitoring, the focus is on monitoring the important electrical parameters, current and voltage, without considering a component level monitoring of the electric traction power supply system [8,9,10]. For example, for equipment such as the impedance bonds, mentioned in [11,12], which play an important role in this system and which are installed on electrified lines to ensure the return circuit of the electric traction current or to separate the signaling current in the circuits of the isolated sections of the sections from the traction current, there is no approach, in terms of monitoring and diagnosis, to reduce the risk of failure of both the equipment itself and the electrical and signaling installations in the electric traction power supply system.

There are two operating regimes in the electric traction power supply system: Normal and fault regime. The fault regime is caused by short circuits on the contact line, in substations posts or in traction substations. Protection of electrical equipment in the power supply system against abnormal operating regimes is provided by various types of specific protection to electrical traction, mentioned in [8,9]. As regards the contact line, it is protected by maximum overcurrent and distance protection.

Protection of railway infrastructure is largely achieved with conventional devices (current relays, voltage relays, impedance relays, etc.) which are outdated and technologically obsolete. In modernized areas of the railway infrastructure, digital relays are used but do not incorporate a large number of parameters for monitoring and decision-making purposes. Thus, basic protection against short circuits occurring in the contact line area is currently achieved by means of distance protection. Distance protection is achieved with complex impedance relays, which control the tripping of circuit breakers when the impedance of the power installation (power supply feeder, contact line, railway tracks and return feeder) decreases below a set value. In order to reduce the time when the contact line is not supplied with electricity, it is necessary, among other things, to identify as quickly as possible where a short circuit has occurred. Impedance relays and digital relays used in the protection of railway installations cannot specify the distance to the fault location and therefore increase the total time to remove the fault condition.

In this sense, some research papers describe some new methods to measure the line impedance in order to improve distance protection relays, in the sense of quick location of a fault, for improving the continuity and reliability of the railway infrastructure [13,14,15,16]. Furthermore, Cho et al. [13] present a fault locator used to detect the fault location with good accuracy.

Moreover, the electro-erosion of the contacts of existing circuit breakers in electrical power installations in railway traction, according to technical data sheets and/or operating instructions, is assessed by counting the number of disconnections (in the normal and fault regime) and the real value of the disconnected current is not taken into account. This leads to maintenance work being carried out at inappropriate times with regard to the electrical wear of the contacts of these circuit breakers [17,18].

In fact, the purpose of the monitoring and protection system is to detect, isolate and locate failures as early as possible. In the context of railway applications and in order to become more performant, safety and reliable, developing effective monitoring and protection techniques is essential from as early as the design phase of the system [19,20].

Fault detection and protection of railway infrastructure is the next step after railway infrastructure condition monitoring. In this sense, some monitoring and diagnosis methods of railway infrastructure have been proposed in the literature [21,22,23]. In [24,25,26], aspects of condition monitoring in the railway industry are presented, in which the sensors used for monitoring the railway infrastructure are highlighted. Moreover, the authors divide the railway infrastructure into two large components: Fixed and movable [24]. The railway tracks, tunnels, railway track beds, bridges and other infrastructure are found in the fixed elements. The track circuit and the return circuit are classified as being part of the other railway infrastructure. On the other hand, the wagon, brakes, axles, bogies, pantographs, etc., are included in the movable elements.

In order to detect some faults in the railway infrastructure, the patent [27] is related to a method and system for detecting a broken track and occupied track from a railway vehicle. The authors of [28] have developed a measurement system for railway vehicles, in the sense of wheel flange wear. Moreover, the authors of [29,30,31] have designed and implemented an online condition monitoring device, which is applicable for track condition monitoring being composed of an on-board device, data collection and remote data collection via cellular phone (GSM). The monitoring device was performed for diagnosis of track fault through measuring the bogie acceleration.

In the last few years, monitoring and protection of railway infrastructure have been based more on wireless sensor networks [4,6,10,24,25,26,27,28,29,30,31,32,33,34,35,36,37,38,39]. Thus, Shah et al. [4] developed a low-cost, wireless and real-time IoT-based sensing system to detect minor fluctuations in vibrations for rail condition monitoring and damage diagnosis. In [6,10,24,25], the authors present new sensor technologies and wireless devices, which can be used in railway systems to assist in decision-making processes for improving maintenance activities through real-time monitoring. In [32], a wireless sensor network is presented for bridge monitoring, part of railway infrastructure. The wireless sensor network collects information (material deformation, temperature, vibration) for knowing the technical condition and behavior of infrastructure when a locomotive moves over. Moreover, Bennett et al. [33] described the wireless network for tunnel monitoring.

Flammini et al. [34] present an early warning system, based on a wireless network, for railway infrastructure monitoring and environmental security monitoring against natural hazards and intentional threats (earthquakes, fire, explosions, etc.). In [35] are shown aspects of the optimal energy resource allocation method of wireless sensor networks for railway status monitoring. Gao et al. [36,37] present wireless sensor nodes with the ZigBee protocol, and self-powered for railway condition monitoring. Buggy et al. [38] present a monitoring system of rail and tramways infrastructure using optical fiber sensors to transmit data. Espinosa et al. [39] describe the design, measurement methodology and experimental validation of an electronic system for monitoring the electrical discontinuity in rails to detect the rail breakage. The system for monitoring is composed of current sensors, an emitter module and two receiver modules to detect electrical discontinuity in rails. Minea et al. [40] present an autonomous/robotic railway vehicle, designed to collect multiple pieces of information on safety and functional parameters of a surface railway and/or subway section, based on data fusion and machine learning.

Moreover, some authors are interested in their research in detecting progressive damage or wear, in real time, of some elements from railway infrastructure which in many cases is not detectable, the layer near the surface of the railway wheels and railway contact strips [41,42,43].

Lamas et al. [44] present a brief review of the evolution of communication technologies. The advantages of the latest generation of broadband communication systems (LTE, 5G, IEEE 802.11ad) and the emergence of wireless sensor networks for the railway infrastructure are also explained.

Thus, in this context of monitoring, diagnosis and protection of installations from the power supply system of electric traction, an innovative integrated solution is proposed for monitoring and protection of electric traction supply installations in order to increase the safety of railway traffic and the safety of the energy supply to consumers, increasing the degree of use of the railway infrastructure, and comprising a monitoring structure of the return circuit from the railway traction, and a monitoring and protection system of the electric traction supply installations that includes the monitoring structures which allow:monitoring of the various parameters from the return circuit and then the processing and transmission data via the GSM network;diagnosing of the return circuit technical condition;determination of the block section, from the structure of the electric traction supply installations, in which an electric fault (short circuit) has occurred;assessment of the electro-erosion of the contacts of some circuit breakers in the electric traction supply installations and an indication of the appropriate moment for intervention concerning them.

Furthermore, the monitoring and protection system will be developed in such a way as to be able to make decisions, based on the input values, when abnormal operating conditions occur (occurrence of a short circuit) in order to protect the railway installation, in the sense of disconnecting the circuit breaker associated with the block section in which the fault occurs.

## 2. Method and Functional Scheme of the Fault Location System from Power Supply System of Railway Infrastructure

The power supply system in electric traction represents the entire set of installation in the sense that the power supply needed to transport people and freight is transformed, transmitted and consumed.

Power supply systems of the electric traction have developed continuously over time, depending on the technologies available at the time. Moreover, the development of this power supply system from electric traction was due to the fact that it allows the supply of traction substations in the three-phase national power system.

The power supply system of electric traction used in our country and in many countries in the world is a single-phase current with a frequency of 50 Hz and a voltage of 25 kV, which is a safe and reliable solution for railway freight and passenger transport [8].

Furthermore, the power supply systems in alternating current electric from electric traction are presented.

### 2.1. The Single-Phase Power Supply System with 50 Hz Industrial Frequency and 2 × 25 kV Voltage with Autotransformers

The single-phase power supply system with industrial frequency and 2 × 25 kV voltage with autotransformers was developed due to the increase in traffic on the railway and the need to remove (reduce) voltage drops that occur when there are more consumers. The application of this power supply system in electric traction was relatively easy, because it can be connected to the 50 Hz, 25 kV single-phase power supply system.

Figure 1 shows the schematic diagram of this power supply system from electric traction. The construction costs are high due to the installation of power autotransformers (ATRs) and the reinforcing feeder, which runs parallel to the contact line, in the supply circuit. The power step-down transformers in the traction substations are connected to all phases of the national power system, and 50 kV voltage is provided in the secondary sense between the reinforcing feeder and the contact line, between which the autotransformers are connected.

The autotransformers are mounted at a distance of 15 km from each other and permanently regulate the voltage on the contact line to 25 kV, regardless of the number of trains in the area. The increasing capacity of rail transport and the increasing distance between traction substations has led to the implementation of 2 × 25 kV and 50 Hz power supply systems in many countries in the world (France, Korea, Italy, Japan, New Zealand, Russia, Hungary, etc.).

The disadvantage of power supply concerning the contact line from the electric traction in the 2 × 25 kV system with autotransformers is the increased cost of the investment in the contact line (feeder, autotransformers, etc.). Therefore, the adoption of a 2 × 25 kV power supply system with autotransformers should only be decided on the basis of a feasibility study compared to direct supply.

Moreover, the power supply systems of electric traction operating in abnormal conditions can affect the operation of other circuits in the vicinity of railway infrastructure, including affecting the human body. In this regard, many papers analyse the disruptive effects of return current from electric traction on signaling circuits.

Thus, when an abnormal return current appears (such as that caused by breakage of one earth conductor or bonding conductor) this can lead to loss of balance and performance of the return circuit, causing in turn increased induction, interference in wayside cables and possible diminishment in tuning and in performance of track circuits [45,46,47,48,49,50,51].

### 2.2. Single-Phase System with Industrial Frequency of 50 Hz and Voltage of 25 kV Connected to Two Phases of the National Power System

This electric traction power system was born out of the need to reduce the cost of the catenary suspension and to be able to direct connection to each country’s national power system. The simplicity of this traction system has made it possible to extend and develop it on all continents because its construction and maintenance are provided at low cost for a high volume of transport and high speeds. The 25 kV voltage and 50 Hz frequency have made it possible to build reliable and easy-to-operate electric locomotives.

Figure 2 shows the schematic diagram of the single-phase 25 kV, 50 Hz power supply system. Electric traction substations (STEs) are supplied from two phases of the power system through 110 kV power lines (LEA).

The step-down power transformers supply the contact line at 25 kV voltage through the power feeder. The traction current returns to the power substation through the railway tracks and return feeders. In our country, in order to increase the volume of freight transported and the safety of railway traffic, the railway track is sectioned into certain lengths called block sections. The traction current is passed from one block section to another by means of special transformers called impedance bonds (simple or double).

Moreover, to ensure the safety of both service personnel and other persons, and to increase the safety of the contact system against short circuit currents, all the railway track constructions that are not normally under voltage are connected; they may be accidentally connected. In general, in this electrical traction supply system all line poles are connected to the railway track by means of a special steel conductor. Moreover, all installations (methane gas, water, heating installations, metal fences, metal bridges, etc.) from the contact line vicinity are connected to the railway tracks to ensure the protection against contact voltage.

The use of the 25 kV single-phase traction system with 50 Hz frequency has the following advantages:electric traction substations are located at long distances (50–70 km) because the voltage on the contact line is high;it is easy to connect to the national power system of 110 kV;construction of traction substations is simple which makes the operation and maintenance costs low;the contact line has a small cross-section (100–150 mm^2^) and low copper consumption;the influence of traction current on underground installations in the vicinity of the track is reduced.

Some weaknesses may also occur in this electric traction power system, such as:the occurrence of inadmissible disturbance voltages in telecommunication lines that are mounted parallel to the railway;when electric locomotives are subjected to high loads over long distances, voltage drops of around 3–5 kV occur on the contact line, which compromises their operation.

Given the advantages of this electric traction power system, it has been developed in many countries, also in Romania.

### 2.3. Method of the Fault Location from Power Supply System of Railway Infrastructure

Furthermore, the following presents the structure of the power supply system from railway infrastructure with the industrial frequency of 50 Hz and voltage of 25 kV, and the method of the fault location when a fault occurs in the power supply system.

Figure 3 shows the detailed structure of an installation of a single-phase power supply system with industrial frequency of 50 Hz and voltage of 25 kV from the electric railway traction. The system includes the power supply (STE—electric traction substation); power supply feeder (FA); contact line/catenary suspension (LC); return circuit (S1, S2—railway tracks; Bj—impedance bond; FÎ—return feeder; Pp—earth connection); and the consumer (electric locomotive). The complexity and diversity of the electrical equipment used in such a system make it necessary to ensure that it is properly operated and maintained in order to avoid abnormal operating conditions which could lead to disturbances in the power supply to consumers. This makes it necessary to monitor, diagnose, remote control and remote signal the power supply system in the electric traction system.

From the point of view of the operating regime, the power supply installations of electrical railway traction may be in the normal regime of operation of the faulty regime. The faulty regime is determined by the occurrence of a defect (short circuit) in the area of the contact line or in the electric traction substation. The basic protection against short circuits occurring in the area of the contact line is currently provided with the help of distance protection. The distance protection is performed with complex impedance relays that control the tripping of the circuit breakers when the impedance of the contact line falls below a set value.

In order to reduce the non-power supply time of the contact line, it is necessary, among other things, to identify as soon as possible the block section where the short circuit occurred. Due to the change of the impedance values of the contact line compared to the impedance values of the normal operating regimes, established when adjusting the protection, the distance to the fault location is not indicated exactly by the impedance relay and thus the total duration of removal of the faulty regime is increased [9]. The value of the contact line impedance changes due to the following causes: Breakdown/electric arc flashover of the insulators in the contact line, and the appearance in the area of the contact line of some conductive objects (vegetation, wires, oversized loads that may enter in the area of influence, defects of locomotives or electric frames, etc.).

Moreover, the assessment of the contact electro-erosion of the circuit breakers from the power supply installations in the railway traction is made by counting the number of disconnections (in normal or faulty regime) and the disconnected real current value is not taken into account. This assessment of the contact electro-erosion is done according to the technical data sheets and/or operating instructions [13]. Finally, this leads to maintenance activities at inadequate times regarding the electrical wear of the contacts of those circuit breakers.

In this context, that of the electric power supply installations in the railway traction, in order to determine exactly the block section where the short circuit occurred, a method is proposed concerning an integrated system that will allow this. The proposed system will also allow knowledge of the real electrical wear of the circuit breaker contacts and will be able to indicate the opportune moment regarding the intervention concerning them.

The method for locating the block section, from the power supply installations in the electric traction, in which a short circuit appeared, supposes that by knowing the return currents through the railway tracks of each block section, the unbalance current between these currents can be established exactly as the block section in which a short circuit occurred.

In the case of the power supply system from the electric traction, the return of the traction current to the electrical traction substation is done through the railway tracks and concerning the return feeder. The railway tracks S1 and S2 are sectioned at certain lengths, called block sections, with insulating joints (JI), in order to control and signal the position of railway vehicles, see Figure 1. The passage of the traction current from one block section to another is done through the impedance bonds, Bj.

Thus, in the return circuit and track circuit among the electrical equipment requiring special attention are the impedance bonds which are installed in the railway infrastructure. The most important capabilities of them are:to carry the return circuit of the traction current, the short circuit currents when they appear;to isolate the track circuit signaling current from the traction current;to be able to signal and control train movements;to protect the installations in the vicinity of the contact line.

A failure of the impedance bond will block the signaling circuit, leading to interruption of the traction power supply system, including the train movement in the block section.

In the case of a faulty regime caused by a short circuit on block section i, see Figure 4, the total short circuit current will be distributed among one or two railway tracks of the faulty block section.

The path of the return currents or the fault currents is through the rail to the electric traction substation where the contact line is supplied. Thus, the short circuit current flows in the left or right sections of the track in the electric traction substation which supplies the contact line. Figure 4 shows the case where the fault arises from the electric traction substation placed on the left side.

The following types of faults are found in a traction supply power system such as the:catenary touching one rail;catenary touching both rails;catenary touching one pole;electrical circuits of the train are breakdown.

The occurrence of a short circuit may be caused by faults in the electrical circuits in the locomotive, broken catenary, breakdown/electric arc flashover of insulation, operating errors (maneuvering /operation) or some random causes.

When an abnormal current in the electrical circuit of the locomotive appears or when the catenary falls on the train body, then current flows through the wheels and railway tracks on the earth connection. In this case, the short circuit currents will be distributed approximatively equally. Moreover, the same consequences happen when the catenary touches both rails.

Another case is when the catenary breaks and hits a pole or one of the rails, and in this cases, the short circuit current will be distributed with a large unbalance between the two railway tracks of the faulted block section.

As a result of the breakdown or flashover of an insulator, the short circuit current closing path from the LC contact line will be through the metal armature of the pole and the steel conductor between the pole and the railway tracks on which the faulty insulator is mounted. The short circuit current closing path will be completed through the railway tracks of the block section where the fault occurred, through the impedance bond from the electric traction substation of that block section, then through the impedance bonds and the railway tracks of the other block sections up to the return feeder.

On the block section where the fault occurred, the impedances of the current paths through the two rails of the short circuit will have different values, resulting in an unbalance distribution of the short circuit current. On the other block sections through the electric traction substation, the distribution of the short circuit current through the railway tracks shall be approximately equal. Depending on how the short circuit is produced (metallic or electrical arc) the current flow through the railway tracks of the block sections after the fault location is very small or even zero.

Therefore, in the case of a short circuit, a high current or high unbalanced currents will flow through the railway tracks of the block section where the fault occurred. Thus, if the return currents through the railway tracks of each block section and the unbalance between these currents are known, the exact block section where a short circuit has occurred can be determined. Even if the unbalance between currents on the faulted block section is not conclusive, by correlating the information from all other block sections it is possible to determine the faulted block section (the last block section with high current values from the electric traction substation).

Therefore, the monitoring and protection system for detecting the faulty block section comprises (see Figure 4):various transducers (current, temperature, oil level, etc.);modules of data acquisition, processing and transmission, MEi;module of reception and signaling for a faulted block section, MR.

The current transducers (TC), mounted on each half winding of the impedance bonds from the traction substation of each block section, monitor the currents through the railway tracks and transmit the information to the modules of data acquisition, processing and transmission. This data is processed at the module level and then transmitted via GSM to the receiver and signaling module for the faulty block section.

Further, it will present the components of the monitoring and protection solution for electrical traction installations.

### 2.4. Module of Data Acquisition, Processing and Transmission

Figure 5 shows the block diagram of the hardware–software structure for monitoring and diagnosing the return circuit. This structure comprises current transducers (TC), temperature transducers (TT) and level transducers (TN); a block of device inputs in accordance with the monitored parameters; a block of relay outputs; a control and information processing unit, microcontroller controlled (UCPI); a data storage module (MSD); a module for configuring and displaying device parameters; RS 232 or USB and wireless communication interfaces (GSM); internal temperature sensor (TINT); power supply unit (BA).

The current transducer, the temperature transducer and the level transducer take the information about the monitored values of the return circuit and apply it to the analog (galvanically separated) inputs of the device.

The information about the monitored currents, via the input block, reaches the data acquisition and processing unit of the module. By analyzing and processing the values of these parameters and by taking into account previous parameter records, we can identify the occurrence of high currents through the railway tracks of the block section and possibly an unbalanced distribution of short circuit current through the tracks.

Through the configuration and display block, the user can set the location of the transmitter module, the characteristics of the transducers used, the threshold values of the monitored parameters and the unbalance thresholds of the currents. At the same time, the transmitter module, through the MSD block, has the possibility of local storage of the values of the monitored parameters when an event occurs (threshold limits exceeded). The transmitter module on each block section transmits the following information remotely to a receiver module, via a GSM modem: Location of the transmitter module (block section); values of the currents through the tracks concerning the unbalance between them; exceeding of threshold values of the currents and of an unbalance current, etc.

In addition to the hardware part of the monitoring and diagnosing structure for the return circuit from the electric traction supply system, there is the software component that involves the appropriate programming of the microcontroller: The reading, processing, and local analysis of the data acquired by the hardware components of the device with the possibility of remote data transmission.

### 2.5. Module of Reception and Signaling of the Faulty Block Section

Figure 6 shows the receiver module which contains: RS 232 and GSM communication interfaces; a microcontroller-controlled data processing unit, UPD; a data storage block, MSD; a configuration, display and alarm block; a power supply block; a computing unit.

The information sent by the transmitter modules, ME, Figure 5, via the GSM modem of the receiver module, reaches the data processing unit of the receiver module, Figure 6. The analysis of the received information determines whether a short circuit has occurred, the value of the short circuit current and the block section where the fault occurred. Through the configuration, display and alarm block, the user can set the number of block sections, the characteristics of each block section (length, short circuit and unbalance current values, concerning threshold current and unbalance values). The block allows the display of information on the faulted block section and the alarm (optical and/or acoustic) on the occurrence of the short circuit. The reception module, via the MSD block, also has the possibility of centralized storage of the values of the monitored parameters of all the block sections at the time of the occurrence of the short circuit.

The reception module, through the data processing block, starting from the values of the disconnected short circuit currents, allows the calculation of the electro-erosion of the circuit breaker contacts in the electrical traction substation. The display and alarm block will show the actual electro-erosion of the circuit breaker contacts, and also indicate when maintenance activity is required on them.

The location of the block section, in the structure of the electric traction supply installations, in which a short circuit has occurred, is characterized in that by knowing the return currents through the railway tracks of each block section and the unbalance between these currents it is possible to determine precisely the block section in which a short circuit has occurred.

In summary, the system for locating the faulty block section from the power supply installations of the electric traction system consists of various transducers; modules for acquiring, processing and transmitting data relating to the currents through the tracks of each block section on which they are mounted, each consisting of input block; a data acquisition and processing unit; a data storage block; a configuration and parameter display block; communication interfaces, GSM communication; a power supply block; a reception module, allowing to locate the faulty block section, composed of RS 232 and GSM communication interfaces; a microcontroller-controlled data processing unit; a data storage block; a configuration, display and alarm block; a power supply block.

The system for locating the faulty block section from the power supply installations in the electric traction system makes it possible, through the system’s receiver module, to know, from the values of the disconnected short circuit currents, the actual contact electro-erosion of the circuit breakers from the electrical traction substation, i.e., when maintenance work on them is necessary.

## 3. Aspects of Monitoring and Protection of Electrical Traction Supply System Installations Using GSM Communication

Monitoring and protection of installations in the electric traction power supply system are beginning to expand due to the development of new technologies. The availability of sensors and transducers, monitoring, diagnosis, remote control, remote signaling and protection devices, and new data transmission (wireless via GSM) possibilities make it possible to create complex monitoring and protection structures that meet the requirements of the power supply system of electric traction.

The main objectives of monitoring, diagnosis, remote control and remote signaling systems of the power system in electric traction are:to acquire information from the monitored power supply system from railway infrastructure;to transmit the acquired data from the power supply system to the central level, wireless via GSM;to know the level of availability of equipment and installations;analysis of equipment faults in the power supply system from railway infrastructure;to process information at central level (railway energy dispatcher);to identify equipment with a high number of failures and interventions;storage of events occurring in the power supply system from railway infrastructure, for post failure analysis;to make decisions when abnormal operating conditions occur (occurrence of a short circuit) in order to protect the railway installation, in the sense of disconnecting the circuit breaker associated with the block section in which the fault occurs;statistical processing of events occurring in the power supply system.

During the maintenance activities of the railway infrastructure, issues related to the return circuit in the electric traction have often been found, namely [5]:insufficient tightening of the connecting terminals;oxidized electrical contacts, broken/missing the connecting ropes between the impedance bond terminals and the railway tracks;broken railway tracks;circuit insulation damage;short circuit in the winding of impedance bond;oil leakage of some equipment (classical impedance bond), etc.

The aspects presented above, and also the occurrence of a short circuit can cause a current unbalance between the two currents on the railway tracks, which can increase above the maximum allowed limit of 10% and can reach up to 100%. This leads to thermal overload of the current paths where these undesirable events occur (conductors, half-windings of the coils at the ends of the block section where the abnormal situation occurred, etc.) concerning the blocking (failure) of the control circuit on that block section.

In this context, to reduce the problems that may occur in the return circuit of the railway traction, it is necessary to monitor the currents through the railway tracks, the temperatures in the contact area, etc. Thus, the monitoring and protection system for railway infrastructure installations offers a solution to some of the issues highlighted, and is focused on the monitoring of impedance bond from return circuit [5,7].

Further, Figure 7 shows an image with the monitoring device of return circuit using GSM communication which monitors the currents through the two railway tracks, the temperatures of some measurement points, considered important from the impedance bond, the oil level and ambient temperature sensor for temperature measurements.

The main elements of the condition monitoring device used to acquire and record the parameters on the return circuit are [5]: ATmega644P microcontroller (Chandler, AZ, USA); two current transducers based on Hall effect of LEM LTS 6-NP type (Geneva, Switzerland); relay outputs; digital temperature sensor of DS18B20 type (San Jose, CA, USA); power supply module of Tracopower TMLM 04105 type (Barr, Switzerland); SD-card module of ROGUE uMMC type for local data storage (Toronto Canada); 2 × 16-character liquid crystal display, for monitoring device configuration and data displaying; communication interfaces with TTL/RS232 converter (Dallas, TX, USA) and TTL/USB converter (Glasgow, Scotland); GSM modem (Adelaide, Australia).

Moreover, the development of mobile phone networks worldwide with increasing performance, the propagation of the Internet and wireless communication techniques make it possible to move many applications from the conceptual level to their practical realization, such as vehicle driving, monitoring, remote control and remote diagnosis of various electrical equipment items and installations, even in the field of railway infrastructure [44,52,53,54,55,56].

Nowadays, there are several communication modes and they can be classified (depending on the data transmission mode) into two main categories: Wired and wireless. The traditional wireless method of data transmission was based on radio waves, which had some disadvantages such as the high price of devices, short operating distance and various interferences with other devices. Thus a new wireless communication technology was adopted, namely GSM.

Modern industrial application control systems require automatic data reading, on-line monitoring and control processing, so remote control is an important feature. One possible solution for remote control is GSM technology, which is available almost anywhere and does not require wiring. The costs of this technology are steadily decreasing due to the increasing number of operators and commercial links between them, making GSM technology very attractive for an increasingly wide range of applications. GSM can be used for data transfer in several ways: Voice, packet data (GPRS—General Packet Radio Switching), data transfer based on Internet technology (TCP/IP—Transmission Control Protocol/Internet Protocol) and last but not least short message service (SMS Short Message Service).

Therefore, an architecture of an integrated monitoring and protection system of power supply system from railway infrastructure using GSM communication is presented below, see Figure 8. The monitoring and protection system contains *n* groups of sensors (current, temperature, oil), *n* monitoring devices, for *n* impedance bonds, and consists of each device equipped with a GSM modem connected to its serial port, a server also equipped with a GSM modem, which receives the information transmitted by the devices, and a software package for processing and analyzing the acquired data and for communication between them.

For measuring the currents through the windings of the impedance bond, a current transformer (inductive type) is used, with a transformation ratio of 250/5 A, the precision class is 1 and the apparent power is 1.5 VA. The temperature sensor used to measure the temperature values on the impedance bond terminals is a Negative Temperature Coefficient (NTC) thermistor of 10 kΩ at 25 °C value with a range of −75 °C to +300 °C, with accuracy ±1%. Some temperature-dependent calibration functions can be used to improve the performance of these sensors by minimizing the effects of temperature variability [5].

Among other things, the monitoring and protection system of power supply system from railway infrastructure allows:knowledge of the currents through the two railway tracks,the temperatures on the contact terminals, on the connecting ropes, on the metallic or non-metallic casing of some of the equipment in the power supply system of the electric traction (impedance bond), the ambient temperature,the oil level (in the case of the conventional impedance bonds);knowledge of the current unbalance on the two railway tracks of the traction supply system;real-time knowledge of the technical condition of some equipment in the railway infrastructure (impedance bond, return circuit, etc.);determination of the block section, from the structure of the electric traction supply installations, in which an electric fault (short circuit) has occurred;assessment of the electro-erosion of the contacts of some circuit breakers in the electric traction supply installations and indication of the appropriate moment for intervention in them;knowledge of the insulation quality of the return circuit.

Furthermore, the monitoring and protection system will be developed in such a way as to be able to make decisions, based on the input values, when abnormal operating conditions occur (occurrence of a short circuit) in order to protect the railway installation, in the sense of disconnecting the circuit breaker associated with the block section in which the fault occurs.

The effort in the building of these monitoring structures is mainly focused on software development, but some measures should also be taken to avoid possible problems with electromagnetic compatibility caused by GSM modems, especially when the antenna is very close to sensitive analog circuits (analog-to-digital converters, circuits needed by the microcontroller for reset or interruptions). Moreover, disturbances produced by the contact line which supplies power to the consumers (electric locomotives) can be a problem for the GSM modem during data transmission.

The implementation of communication protocols in an integrated system used for monitoring and diagnosing components in the return circuit (impedance bonds) of electric traction is not an easy task due to limited memory resources.

GSM communication is a complex process that requires highly developed infrastructure. During communication between two mobile terminals, the signal passes through many devices (mobile stations, fixed stations, repeaters) and each of these adds its specific delays. This results in a GSM communication time of several hundred milliseconds.

Taking into account the characteristics of a system for monitoring, diagnosing and acquiring data from railway infrastructure, the most suitable data communication mode is GSM, which allows data transmission in an extended system over a large geographical area.

GSM modems are used as the interface between the monitored equipment from the return circuit of the traction power supply system, the GSM network and the receiver module. The GSM modem used in this application is OSTENT Q2303, produced by Wavecom (Adelaide, Australia) [57]. It contains in addition to the power supply part, an antenna and an interface for serial communication type DB9. GSM modems allow wireless communication of text messages, data, voice and fax transmission.

A GSM modem was used in the return circuit monitoring and diagnostic device for the transmission of monitored data (current, temperature, etc.). The modem is connected to the proposed hardware/software structure via RS232 serial interface and controlled by AT (Attention Command) commands [58]. This set of commands was introduced by Hayes and has established itself as a standard GSM modem command language. Before using the GSM modem for data transmission, it is necessary to configure it, test the GSM signal and determine the data format. Message decoding is done through software within the ATMega644P microcontroller equipped to the hardware/software structure.

The second GSM modem can be connected directly to a computer or via the receiver module and it can transmit or receive data. This will transmit commands and receive data to be processed by the computer. The GSM modem allows messages to be received in two ways via the central Short Message Service Center (SMSC) service. The first way is to receive data directly via the serial interface, but requires an online computer or the data will be lost. The second way of receiving data is using the SIM card and a memory card (from the receiver module) to save data. As long as the computer is not available, the received data will be stored on the device’s card and read when the modem is connected to the computer.

The device’s role is to prevent possible faults in the operation of the railway signaling installations on the electrified lines in the traction power system, concerning the avoidance of the return circuit interruption that will lead to delays in the circulation of the trains, and some technical and economic loses.

## 4. Testing the Monitoring and Protection System in Real Conditions from Railway Infrastructure

The experimental tests were performed in order to determine the correct working of the monitoring device under real working conditions specific to the electric traction system.

With the device developed, after laboratory tests of the return circuit [5], the field verification was carried out, where test conditions required the device to be mounted in a relay cabinet near the railway infrastructure. To install the device in the field, two current transformers (TC1 and TC2 with 250/5 A transformer ratio) were mounted on the connecting conductors between the impedance bond and the railway tracks, as well as four NTC type temperature transducers on the terminals and its housing, Figure 9.

To measure the currents through the windings of the impedance bond, a current transformer (inductive type) is used, with the transformation ratio of 250/5 A. Each transformer is designed to be mounted in measuring circuits without the need to undo the connections of the connecting conductors between the terminals of the impedance bond and railway tracks. For current measurements, the precision class is 1, and the apparent power is 1.5 VA [5].

In order to interface the current transformer for measuring high currents, within the device, two inputs characterize the current transducers. Current transducers (with Hall Effect) are placed on the main board of the device. They have a rated current of 6A and can measure the alternating current [5].

The temperature sensors used to measure the temperature on some points from the return circuit are of the Negative Temperature Coefficient (NTC) thermistor type. Generally, they are made of metal oxides (manganese, nickel, cobalt, copper, iron, etc.) and have a temperature measurement range of –75 °C to +300 °C, with accuracy ±1% [5].

The chosen thermistors have nominal values of 10 kΩ at 25 °C and are interfaced to the microcontroller via a voltage divider network, which is designed so that the power dissipated on the thermistor never exceeds 1 mW (in reality, this power is even lower) [5].

By using all of these sensors described above, the return circuit parameters are acquired. These measurements provide valuable information about the return circuit and allow, in most cases, accuracy in checking the presence or absence of anomalies [5].

The four temperature transducers used TT_A1_, TT_A2_, TT_M_ and TT_C_ are mounted on the connection terminals A1, A2, M on the impedance bond, and are used to monitor the temperature at the contact areas and the housing. They are connected to the impedance bond terminals and the housing by means of copper connectors, see Figure 10, which are fitted through clamping screws to the impedance bond. All the outputs from the temperature transducers are inserted into a protective tube and taken to the device, which is located in the relay cabinet. To protect the current transformers from environmental conditions (rain, grid, dust, etc.), they were mounted in a plastic protective housing. From each current transformer, the secondary electrical conductors were mounted in a protective tube. The distance between the monitored impedance bond and the cabinet where the device was mounted was about 8 m.

Once the current and temperature transducers were installed on the monitored impedance bond, they were connected to the monitoring device located in the relay cabinet, see Figure 11. The GSM module was also mounted next to the device for data transmission during the monitoring of the impedance bond parameters.

The device was also configured by entering the kilometric position of the block section where the impedance bond is mounted, the minimum and maximum temperature values for each temperature transducer (e.g., −20 °C and +80 °C), at what value of the current through the railway tracks the acquisition should start, the duration of the acquisition and from what value of the current unbalance the alarms should be issued. After mounting and setting up the device, the impedance parameters were monitored and the parameter values were transmitted to a central server once the threshold value was exceeded. Data acquisition in the file is stopped when the maximum number of data with values is reached after which a new file will be created, i.e., when the acquisition is stopped.

For example, Figure 12 shows the values of the monitored parameters (currents through the two railway tracks) in a file for a given time interval. In Figure 13 a 12-min sample is extracted, from which the evolution of currents and temperatures on the terminals of the impedance bond can be observed. The monitored currents in the return circuit with the measuring device had approximately 42 Arms value, given by the load from the circuit.

In the normal operation of the power supply system from electric traction, the total traction current, I_LC_, will be approximately equally distributed on the two railway tracks (IS_1_ ≈ IS_2_ ≈ 0.5I_LC_). This means that the total traction current will flow through the median of the impedance bond.

Moreover, it can be seen that the traction currents through the railway tracks had a small unbalance during the 800 s of recording, see Figure 13a. Thus, an unbalance current of about 3 A occurs between the two track rails, mainly due to the connections in the return circuit of the respective rail.

The temperature evolutions on the three terminals of the impedance bond in the presence of these currents are given in Figure 13b. By analyzing the evolution of the temperatures on each contact zone, the over temperatures on these for 800 s can be determined. The temperature value on the middle terminal (3) of the impedance bond is higher because it is transited by the currents on the two railway tracks. Moreover, the temperature values on terminal 2 of the impedance bond have a higher rise than those on terminal 1. Thus, at the end of the 800 s duration, the temperature on terminal 2 is 0.25 °C higher than that on terminal 1. This is due to a higher contact resistance on terminal 2, which requires maintenance work on this connection.

## 5. Conclusions

Using an integrated solution for monitoring and protection of power supply system of electric traction with several sensors, monitoring devices mounted on railway infrastructure for collection and analysis of data in the field and data transmission via GSM to a central dispatcher can ease the work of human operators. The monitoring and protection solution will allow to know exactly which block section of the power supply installations of railway traction is affected by an electrical fault (short circuit) and to alert the operator with warning messages, at a central/dispatcher level. Thus, the time to detect the block section where the short circuit occurred is very small, and the human operators will know exactly where to go to carry out the maintenance activities.

Depending on the disconnected short circuit current, it shall cause electro-erosion of the contacts of circuit breakers from the electrical power supply installations of the electric traction and indicate when it is appropriate to intervene in them.

Moreover, the monitoring and protection solution will allow the monitoring of parameters, with the monitoring device of the return circuit of electric traction, such as currents through the two railways tracks, temperatures, oil level, type of monitored equipment, quality of the current return circuit, quality of the infrastructure insulation, characteristics of the transducers used and threshold values of the monitored parameters. It can detect various conditions in which the return circuit installation may be: Insufficient tightening of some connecting terminals, broken (sometimes missing) ropes between the impedance bond and railway tracks, broken railway tracks, short circuit in the winding of impedance bond, etc.

Last but not least, the monitoring and protection system will be able to be developed in such a way as to be able to make decisions on the basis of the input values when abnormal operating conditions occur (occurrence of a short circuit) in order to protect the railway installation by disconnecting the circuit-breaker associated with the block section in which the fault occurs.

In conclusion, from a scientific and technical point of view, the solution for monitoring and protection of the railway infrastructure will be a concrete means of introducing technological progress, with elements of originality, with the consequences of increasing the safety of the energy supply to consumers, the energy efficiency of the installations within the power supply system of the electric traction, increasing the safety of railway traffic and the possibility of increasing the degree of use of the railway infrastructure.

The experimental tests performed in the field on the railway infrastructure tested with various sensors showed good functioning concerning the monitoring and protection system of power supply system from railway infrastructure, in the sense of monitoring of return circuit parameters and then transmitted data GSM modem to a central server where the data can be analyzed and interpreted. Thus, the monitoring and protection system fulfilled the purpose for which it was designed and realized. On the other hand, the high temperature from one measured point can give us a fault that appears in the contact area and requires maintenance work on this connection.

Based on the present achievements, future research will focus on developing the software application for an easy interface with the operator.

## 6. Patents

Adam M., Munteanu A., Pancu C.M., Andrusca M., “Method and system for locating the faulty block section in power supply installations in railway traction”, no. a 00936, 2018, Romania. Adam M., Munteanu A., Pancu C.M., Andrusca M., “Method and apparatus for monitoring and diagnosis of impedance bond”, Request patent no. a 00233, 2017, Romania.

## Figures and Tables

**Figure 1 sensors-21-07858-f001:**
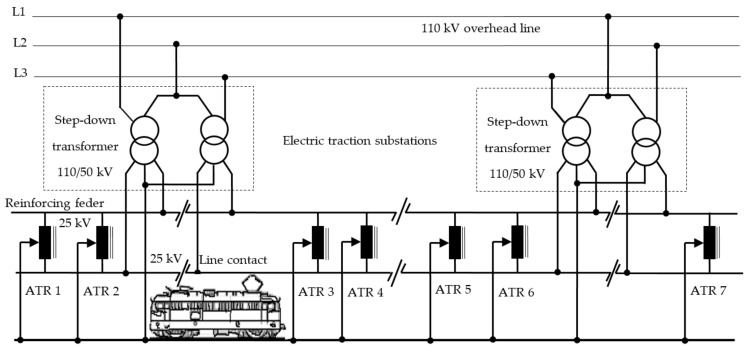
Schematic diagram of single-phase system with 25 kV, and 50 Hz with autotransformers.

**Figure 2 sensors-21-07858-f002:**
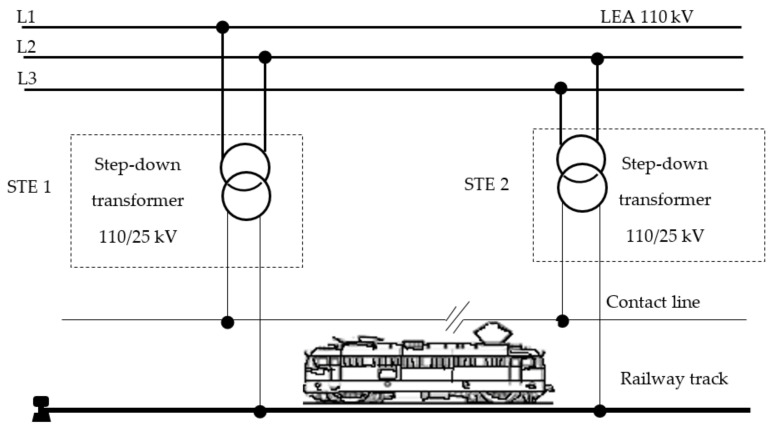
Schematic diagram of single-phase system with 25 kV, and 50 Hz.

**Figure 3 sensors-21-07858-f003:**
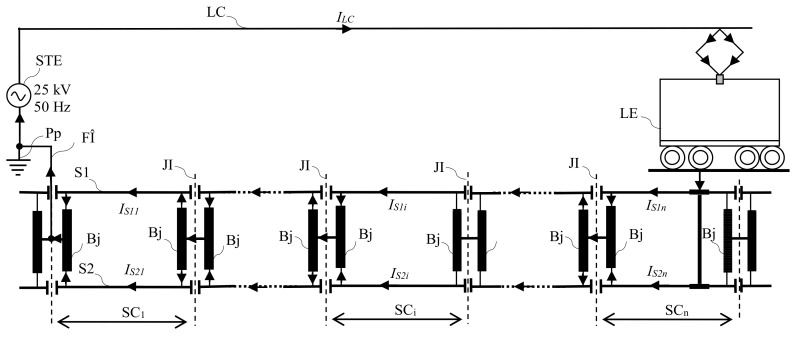
The structure of a power supply system from railway infrastructure.

**Figure 4 sensors-21-07858-f004:**
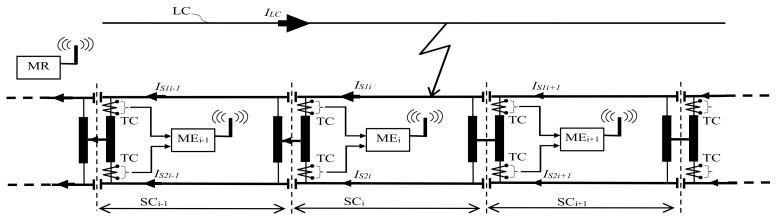
Functional block diagram of the fault location system.

**Figure 5 sensors-21-07858-f005:**
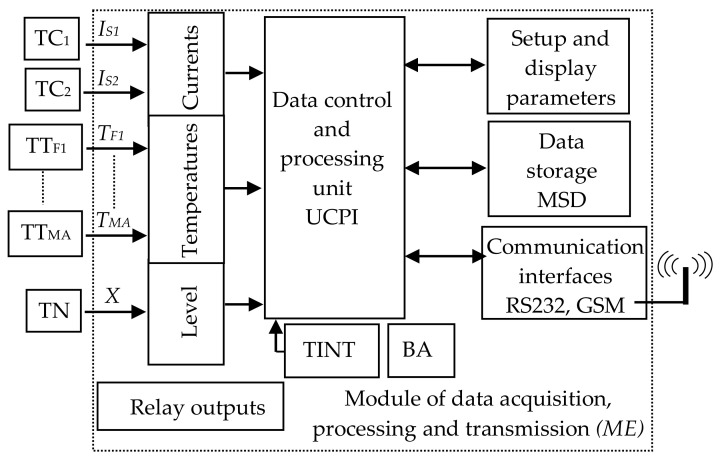
Block diagram of the monitoring device of the return circuit from the electric traction power supply system.

**Figure 6 sensors-21-07858-f006:**
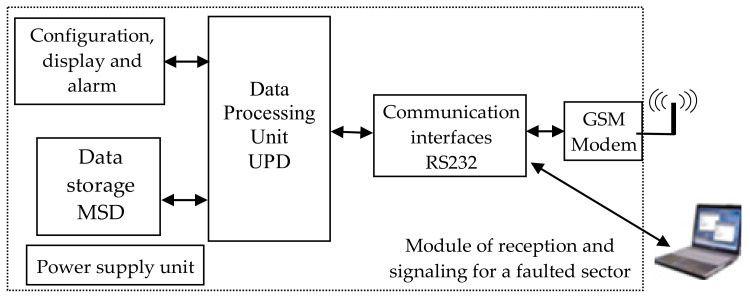
Schematic diagram of the module of reception and signaling of the faulty block section.

**Figure 7 sensors-21-07858-f007:**
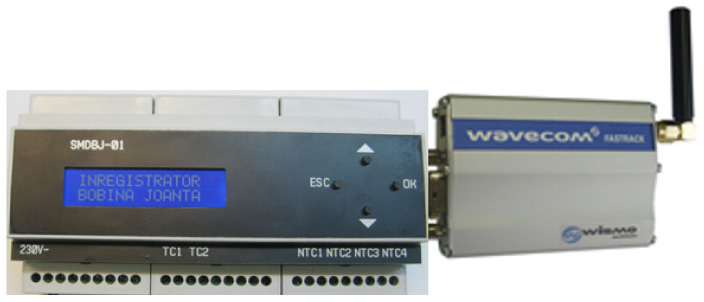
Monitoring device of return circuit using GSM communication.

**Figure 8 sensors-21-07858-f008:**
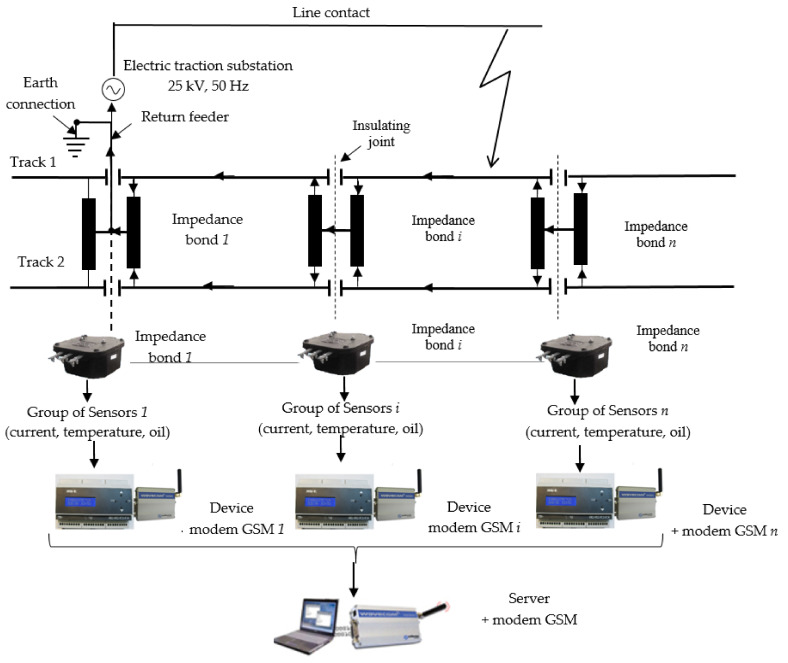
Architecture of an integrated monitoring and protection system of power supply system from railway infrastructure using GSM communication.

**Figure 9 sensors-21-07858-f009:**
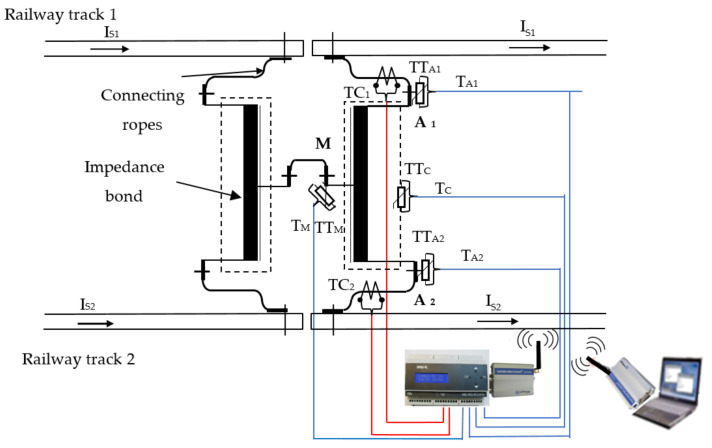
Field monitoring wiring diagram of return circuit.

**Figure 10 sensors-21-07858-f010:**
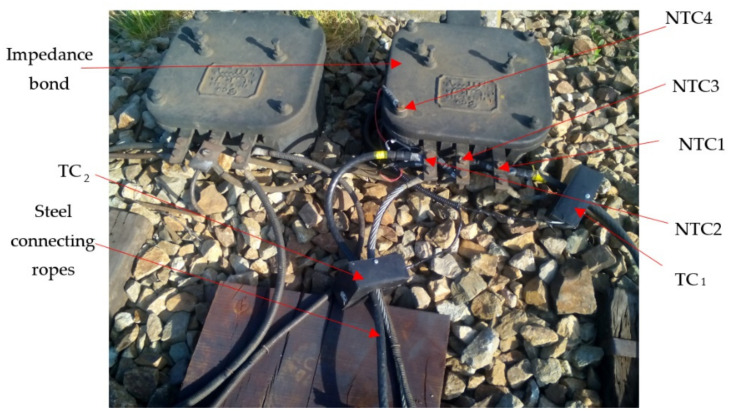
Image of the current and temperature transducers mounted in return circuit of oil impedance bond in the field.

**Figure 11 sensors-21-07858-f011:**
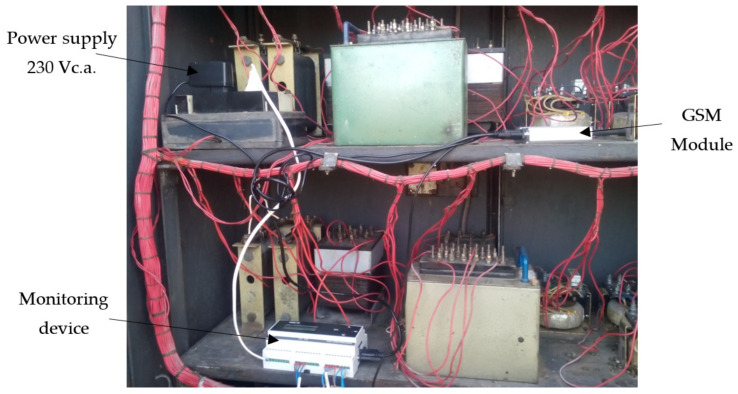
Image of the monitoring device in the relay cabinet from the field.

**Figure 12 sensors-21-07858-f012:**
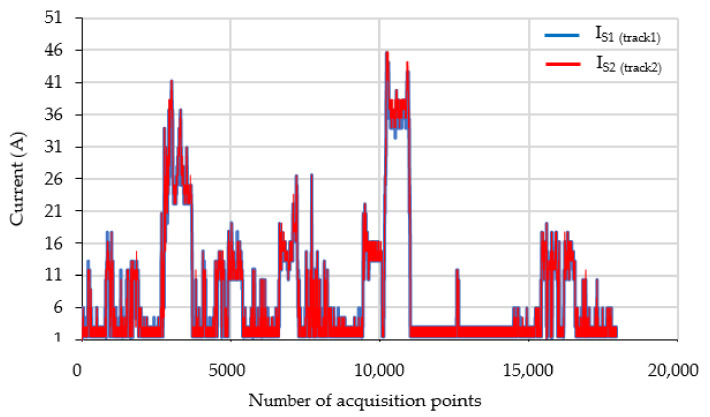
Monitoring the current from return circuit over a period of time.

**Figure 13 sensors-21-07858-f013:**
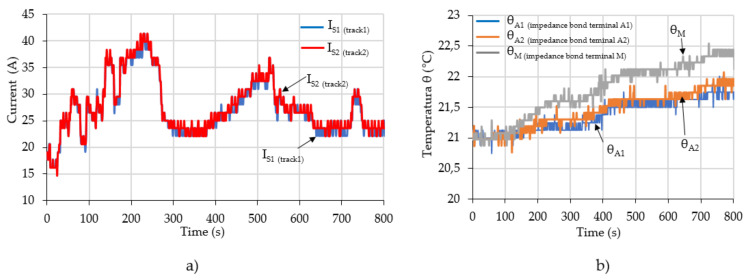
Monitoring the return circuit: (**a**) Currents evolution; (**b**) temperatures evolution.

## Data Availability

Not applicable.

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
