# Peer review of "Innovative Integrated Solution for Monitoring and Protection of Power Supply System from Railway Infrastructure"

_sensors, 2021, doi:10.3390/s21237858_

Round 1
Reviewer 1 Report
The work my be considered as composed of two parts: one regarding the fault scenarios and the operation of the railway system, justifying the quantities subject to measurement; one describing the system applied for the measurement, data acquisition, processing and transmission.
(General comment) The first part is not considered in detail, as it should be clarified which type of fault, which effectiveness is expected, why existing methods fall short or deserve improvement, with the due references to the literature. For the second part it may be acceptable a high-level description of the communication system, but the way the data are processed to get the necessary information and to implement the necessary actions must be detailed. Measured quantities should be linked to phenomena to monitor with physical explanation (first part), thus setting thresholds for detection, criteria, etc.
0) Regarding sensoring of rails and interconnection of sensors, i see that you have not considered a recent publication in Sensors:
Ref0: "Towards the Internet of Smart Trains: A Review on Industrial IoT-Connected Railways", 10.3390/s17061457.
1) Section 2. You focus on the objective of fault location, but this is a tricky point:
(i) existing protections and distance relays already do a good job;
(ii) there are publications discussing methods for fault detection;
(iii) in case of insulator breakage you cannot see the return current with your method because it flows back directly to earth, through the mast;
(iv) all in all you can detect falling catenary and derailed pantograph, that absolutely are such big mishaps that are clearly visible; probably there is a train trapped somewhere and passengers leaving the train. (it may be worth remembering that often the phone call of a passenger to the local emergency number anticipates the communication of the infrastructure owner).
for (ii) for example you should consider recent publications, such as:
Ref.1: "A Novel Fault-Location Algorithm for AC Parallel Autotransformer Feeding System", 10.1109/TPWRD.2018.2872165 ;
Ref2: "Research of Implementing Least Squares in Digital Distance Relaying for A.C. Electrified Railway", 2004 ;
Ref3; "Line Impedance Measurement to Improve Power Systems Protection of the Gautrain 25 kV Autotransformer Traction Power Supply System", 10.1109/ACCESS.2019.2940894 ;
Ref4 (old reference, but good overview): "AC ELECTRIFIED RAILWAYS - PROTECTION AND DISTANCE TO FAULT MEASUREMENT", 1995
This said, I would suggest that you elaborate more your approach in order to extend it to situations of abnormal return current division (such as caused by breakage of one earth conductor or bonding conductor) with benefit not for the detection of a significant short-circuit event, but for loss of balance and performance of the return circuit, causing in turn (a) increased induction and interference to wayside cables, (b) possible diminished tuning and performance of track circuits.
For (a) you can refer to the following references:
- Ref5: "Influencing Factors of the Induced Voltage in Signal Cable of Electrified Railway", 2009 ;
- Ref6: "Induced Voltage Calculation in Electric Traction Systems: Simplified Methods, Screening Factors and Accuracy", 2011 ;
- Ref7: "Influence of railway line characteristics in inductive interference on railway track circuits", 10.1049/iet-smt.2018.5021.
For (b) you can refer to the following references:
- Ref8: "Development and performance analysis of a novel impedance bond for railway track circuits", 10.1049/iet-est.2013.0004
- Ref9: "Modeling of Audiofrequency Track Circuits for validation, tuning and conducted interference prediction", 10.1109/TITS.2009.2029393
2) Line 214-218. The sentences here are generic and should be backed up by references. E.g. distance relays are not accurate enough for positioning of fault; but if you measure only the quantities at the impedance bonds that are often located at 1-1.5 km separation, then you are even less accurate!
3) Figure 5. You have not clarified the principle by which you monitor and detect fault events. In Figure 5 you show impedance bonds, that hints that this is the proposed method (from a system and electric viewpoint, not from an implementation and communication viewpoint).
Well, there is no reason that the track current is less balanced during a short circuit rather than during normal operation. (on balance see other comment) Then, you just measure the return current and the rail current at the impedance bonds, that is possible only if the return current passes through the impedance bonds. For a 50 Hz system this is a peculiar configuration (the one shown in Figure 2), that is not commonly applied in 25 kV 50 Hz. The usual impedance bond is just 1 winding between rails and a center tapping point connected to earth return cables. In this case you can still measure the unbalanced current, but not the total current.
Could you please elaborate on this providing details of the measurement setup and approach?
4) Line 569. If the absorbed current by the running rolling stock is not reported, it is not possible to say that the rail currents are balanced. However, data appearing in the literature (see refs below) confirm a 6 to 9% unbalance, that used to extrapolate from results of Figure 10 gives a total maximum traction current through the track of 500-650 Apk (350-450 Arms approx.). This is more or less the maximum current absorbed by a high-speed train, or two medium-speed trains. Please, provide an estimate to better concretize the data in Figure 10.
- Ref10: as Ref9
- Ref11: "Modelling of the Distribution of Return Traction Current Harmonics in Electrically Asymmetric Rails", 10.1109/EMCEUROPE48519.2020.9245767
5) References are not in the correct format and have typing mistakes (e.g. MPDI for MDPI). There are asterisks for the authors, or normative body, or else.
Concluding remark. At this stage this work is just a proposal of a distributed measuring system that might have interesting applications, but at the moment does not demonstrate its effectiveness at least for the task it is proposed, fault location and in general diagnostic of traction supply.
Reviewer 2 Report
The authors presented a critical topic on monitoring and protection of power supply systems for railway infrastructure. The paper is well-written and illustrated. The following comments should be addressed:
- Although the power supply system for railway infrastructure is critical in operations, the reviewer suggests adding related content for monitoring wholesome railway infrastructure in the introduction section.
- Although Structurally, the progressive damage in railway infrastructure is not detectable in some cases, however, it is presumably might not be the case with power supply, how do authors think is tackled in power supply systems?
- Did the authors look at the effect of temperature variability in sensors?
- The authors are suggested to look at the progressive damage and cite the following papers
(a) Donzella, G., Faccoli, M., Mazzù, A., Petrogalli, C., & Roberti, R. (2011). Progressive damage assessment in the near-surface layer of railway wheel–rail couple under cyclic contact. Wear, 271(1), 408–416.
(b) Judek, S., & Skibicki, J. (2018). Visual method for detecting critical damage in railway contact strips. Meas. Sci. Technol., 29(5), 055102.
(c) Sony, S & Sadhu, A. (2019). “Identification of progressive damage in structures using time-frequency analysis”, CSCE General Conference, Montreal, Canada.
(d) Noertker, J. A. (1933). Improved power supply betters street railway service. Electrical Engineering, 52(1), 22–28.
Round 2
Reviewer 1 Report
Dear Authors, thank you for your clarifications and the revision of the paper.
For this new version I have comments that partially were already included in the review of the first version, but were mixed to other more important things. In general the paper has a problem of flow of description: sometimes sentences are not in the right place, a topic is split in different thread of discussion in different sections and subsections, or figures are less complete than it should be advisable.
1) Line 333. To make it clear what a circulation sector is when used for the first time you should anticipate this description. In addition, the best way of calling it to my knowledge is "block section", where block is from the signaling jargon and you can distinguish fixed blocks (e.g. implemented as in the present case with track circuits) or moving blocks (e.g. by means of ERTMS, or CBTC in metros).
2) Line 337. When speaking of short circuit please, underline that you assume a falling catenary, touching one of the rails. You speak of unbalance (and for this you need that the catenary touches one rail). But you could speak of abnormal current, and in this case you get also short circuits with catenary falling on the train body and then current flowing through the wheels.
Another case is the catenary breaking and hitting a mast (aka pole) and then short circuiting to the return circuit, because you said that these poles are bonded to the return circuit. So, you could illustrate also this case, that is covered if you monitor the total current intensity.
Please, add a bullet list or table of these cases (catenary touching 1 rail, touching both rails, touching one pole) and indicate briefly which setting/arrangement of your system can detect it.
3) Line 353 & Figure 4. You mean rails, not tracks, because you speak of imbalance. Of course you must consider that the short-circuit current flows in the left and right sections of the track at the point of fault.
In Figure 4 you indicate instead the currents flowing in one direction. This may happen if the system is 1x25 kV and supplied by one end with the other end left floating. That's ok, but please, for generality, show that your system is able to consider current flowing in all directions and can process it.
4) Line 369. Oil level is mentioned here for the first time (then it is clarified better, but after some pages). In general you should clarify which quantities are monitored and for which reason: so, if your system was created for monitoring the health status of the impedance bonds, and if the I.B. happen to have an oil insulated transformer, then you say that you monitor the oil level.
Since the emphasis and focus is so short-circuit detection, I think that the oil level is of no use s input quantity for that (and the temperature too, unless you correct for the I.B. parameters). It is ok to list the various capabilities of your hardware, but you should underline that some quantities are for other purposes and not chiefly for the short circuit under discussion.
5) Line 493-502. Maybe this sentences you should be anticipated or at least summaried where Figure 4 is and where you show the schematic of the return circuit. The consequence of current imbalance is not only overheating of the IB but also false occupation or bad operation of the track circuits.
6) Line 503-504. Sentence does not flow: "in order to ....", then "problems that ...."
At line 505 you speak generally of "some equipment" and "possibly oil level": please, clarify the quantities and what is their use, that means this class of track equipment has these problems that can be tackled with the e quantities, this other class with these other quantities, etc.
7) Line 696. Effective values : you should say rms values.
8) Figure 13. Be careful that units of measure are both between [] and () in the same figure. MDPI requires the use of ().
9) Communication part and overall system. It would be interesting to report performance of data transmission and indicate the capabilities in this sense. The part dealing with communication is only descriptive.
10) Section 4, figures 12-13. You should comment more extensively the results, such as quantifying more accurately the current imbalance, and justifying the visible quantization noise. Also you should indicate the estimate of the short circuit position, clarifying the method used to identify the short circuit and its effectiveness.
11) General. The paper should be rearranged with a more ordered exposition and the speech flowing better from topic to topic. For instance Section 3 is quite long and considers different things but has no subsections arranging things in a more ordered way. The architecture in Figure 8 should be possibly anticipated. Section 4 with results should be increased adding more results and some comments, as already said in the comments of rev1 ; section 4 should be more informative.
As a final consideration, please, check your English form, not for grammatical errors, rather for syntax and form of expression. often sentences are more complicate than necessary, or the speech does not flow as it should be.
Author Response
We would like to appreciate your time and efforts in reviewing this paper. Also, we would like to greatly thank you once again for all your valuable and constructive comments. We have revised the manuscript in response to the thoughtful and constructive suggestions of your comments. We hope our revision has improved the paper quality to a level of satisfaction. Our point-by-point responses to your comments are presented in the attachment.

Reviewer 2 Report
The authors have addressed all my comments, I recommend the article for publication.
Author Response
We would like to greatly thank you for all your constructive and valuable comments and your interest in this paper. Your feedback helped us to improve the quality of the manuscript. Also, we would like to appreciate your time and efforts in reviewing this paper.
Round 3
Reviewer 1 Report
Dear Authors,
thank you for your further improvement and your kind replies.
I have no other remarks.